# LEARNING BOOLEAN FUNCTIONS WITH NEURAL NETWORKS

## ABSTRACT

Many works have shown learnability of functions on the Boolean hypercube via gradient descent. These analyses of gradient descent use the convexity of the problem to establish guarantees despite the fact that most loss functions are highly non-convex. In addition, the analyses explicitly show that the hypothesis class can approximate the target function; this is known as a representation theorem. In this work we give gradient descent guarantees for learning functions on the Boolean hypercube on both the mean squared and hinge losses with 2-layer neural networks with a single hidden non-linear layer. Furthermore, all of our analyses apply to the ReLU activation function. Moreover, on both losses, we don't make use of any convexity of the problem, and don't explicitly prove a representation theorem. A representation theorem is a consequence of our analysis. In the hinge loss setting to learn size $k$ parities, with dimension $n$, and $\epsilon$ error, we obtain bounds of $n^{O(k)}poly(\frac{1}{\epsilon})$ and $n^{O(k)}\log(\frac{1}{\epsilon})$ for network width and samples, and iterations needed, respectively. This upper bound matches the SQ lower bounds of $n^{\Omega(k)}$. In the mean squared loss setting, given that the Fourier spectrum of an activation function has non-zero Fourier coefficients up to degree $k$, and given that the best degree $k$ polynomial approximation of the target function is $\epsilon_0$ in mean squared loss, we give guarantees for network width and samples, and iterations needed of $n^{O(k)}poly(\frac{1}{\epsilon})$ and $n^{O(k)}\log(\frac{1}{\epsilon})$ respectively for an error of $\epsilon + \epsilon_0$. To the best of our knowledge, our bounds of $n^{O(k)}\log(\frac{1}{\epsilon})$ iterations needed for learning degree $k$ polynomials on both losses are better than any previous bounds in the Boolean setting, which is a consequence of not using any convexity of the problem in our analysis. Specifically, in other works in the Boolean setting, the bound on iterations is $n^{O(k)}poly(\frac{1}{\epsilon})$. Moreover, as a corollary to our agnostic learning guarantee, we establish that lower degree Fourier components are learned before higher degree ones, a phenomenon observed experimentally. Finally, as a corollary to our mean squared loss guarantee, we show that neural networks with sparse hidden ReLU units as target functions can be efficiently learned with gradient descent.

## 1 INTRODUCTION

In recent years deep learning has been successful on a variety of practical tasks such as computer vision Krizhevsky et al. (2012). Despite this success, less is known about why neural networks are able to perform so well on these tasks. Moreover, there is a large gap between a theoretical understanding of common training algorithms, such as gradient descent, and what these tools are able to accomplish in practice. Since real world data can be difficult to describe from a theoretical perspective, it makes sense to first understand the training process of neural networks under more well-understood data distributions such as standard Gaussian or uniform Boolean input distributions (Andoni et al., 2014; Daniely, 2017; Du et al., 2018; Malach & Shalev-Shwartz, 2020; Yehudai & Ohad, 2020; Barak et al., 2022; Damian et al., 2022).

Training guarantees in more well-understood settings have advanced our understanding of deep learning, but many of these works have unreasonable model and training assumptions. For example, in many works, assumptions on learning rate schedules (Daniely & Malach, 2020; Barak et al., 2022), weight initialization (Daniely & Malach, 2020; Malach & Shalev-Shwartz, 2020), bias terms (Malach & Shalev-Shwartz, 2020; Barak et al., 2022), and activation functions (Andoni et al., 2014;

Daniely & Malach, 2020) are often unrealistic compared to what is standard in practice. Furthermore, it is often the case that these works have analyses that make use of convexity results despite the fact that most losses are highly non-convex (Daniely, 2017; Yehudai & Shamir, 2019; Daniely & Malach, 2020; Barak et al., 2022). Thus we would hope any analysis would steer away from any convexity in the problem, especially if we wish to extend results to non-convex settings.

In this work we give efficient, in time, model and sample size, agnostic upper bounds for neural network training with gradient descent for learning any function on the Boolean hypercube with the mean squared loss. Specifically, we show that neural network training with gradient descent converges to the best degree $k$ polynomial approximation for the target with network width, samples, and iterations needed of $n^{O(k)}$. In addition, we give $n^{O(k)}$ upper bounds, in samples, width, and iterations for learning $k$ bit parities on the hinge loss. The upper bounds for learning parities on both the hinge and mean square loss match the Statistical Query (SQ) lower bounds of $n^{\Omega(k)}$, noting that gradient descent is a SQ algorithm. Thus, compared to any SQ algorithm, gradient descent is the best we can do for learning parities. Our network model assumptions are reasonable, and our analysis applies to a wide range of activation functions with ReLU as a special case. In the mean squared loss setting, the techniques we employ are similar to those of Vempala & Wilmes (2019), and avoid any convexity of the problem despite the fact that we update only the output layer weights. The avoidance of convexity allows us to achieve upper bounds on iterations of gradient descent of $\log(\frac{1}{\epsilon})$ compared to $poly(\frac{1}{\epsilon})$ for works that take convexity into consideration. To the best of our knowledge, our bounds on iterations are better than any previous work in the Boolean setting. In addition, as a corollary in the mean squared loss setting, we show that neural networks with sparse hidden ReLU units as target functions can be efficiently learned with gradient descent. Finally, in the mean squared loss setting, the analysis gives a spectral bias result for gradient descent that shows low-frequency Fourier coefficients are learned before high-frequency ones.

## 2 RELATED WORK

There are various works that give guarantees for gradient descent when training overparameterized neural networks in the discrete setting on a finite number of input and label pairs. In this setting there is usually some requirement on the input data, and gradient descent iterations are polynomial in the number of samples. For example in (Du et al., 2018; Arora et al., 2019; Song & Yang, 2019) iterations needed and network width are inversely proportional to the smallest eigenvalue of a matrix such that the matrix is a function of the input data. In (Li & Liang, 2018; Allen-Zhu et al., 2019; Zou et al., 2020; Oymak & Soltanolkotabi, 2020; Ji & Telgarsky, 2019) iterations needed and width are inversely proportional to the minimal distance of input data pairs.

In contrast to the discrete setting, there have been many works that allow uncountable distributions (Andoni et al., 2014; Brutzkus & Globerson, 2017; Daniely, 2017; Ge et al., 2017; Li & Yuan, 2017; Soltanolkotabi, 2017; Goel et al., 2018; Damian et al., 2022; Du et al., 2018; Vempala & Wilmes, 2019; Yehudai & Ohad, 2020; Xu & Du, 2023). Even though the hypercube is a countable distribution, the techniques in the discrete and overparamaterized setting, mentioned above, won't work since there are an exponential number of Boolean vectors. Although the hypercube is countable, our analysis in the mean squared loss setting is similar to the one in Vempala & Wilmes (2019) on the sphere. Since the only facts about the sphere they use are that harmonic polynomials form an orthonormal basis for $L^2(\mathbb{S}^{n-1})$ and are eigenfunctions of Funk transforms, we can use a similar analysis since the parities are an orthonormal basis for $L^2(\{-1,1\}^n)$ and are eigenfunctions of the operator of Proposition 4.1.

A large number of papers have given gradient descent training guarantees for learning parities with the hinge loss on the hypercube, or in more general settings, on the uniform and other distributions (Daniely, 2017; Yehudai & Shamir, 2019; Malach & Shalev-Shwartz, 2020; Barak et al., 2022; Shi et al., 2022; Abbe et al., 2023). Most of these works show explicitly that by randomization, the neural network has the capacity to approximate the parity in question at initialization Malach & Shalev-Shwartz (2020) or after the first iteration of training (Daniely & Malach, 2020; Barak et al., 2022); with the approximation result, it can then be shown that gradient descent reaches a good solution using the convexity of the problem (See Theorem 11 of Malach & Shalev-Shwartz (2020)). In addition, the gradient descent analyses in these works and other settings are dependent on these existence results, often with long and complicated proofs (Andoni et al., 2014; Daniely,

2017; Barron, 1993). In contrast, for both losses in our work, the analysis of gradient descent is independent of a representation theorem, a representation theorem is a consequence of our analysis. Our analysis happens in the opposite order: we carry out the analysis of gradient descent on both losses, and our upper-bounds show that the network must be a good approximation of the target function. With both losses we are able to get upper bounds in iterations of gradient descent that are logarithmic in the error as opposed to polynomial in the error for convex techniques. To the best of our knowledge, these logarithmic error bounds on both losses are better than any bound on iterations, for any previous work in the Boolean setting.

In the hypercube setting with the hinge loss, the work of Barak et al. (2022) is the most similar to ours. The work shows that ReLU activation functions with Boolean weights have Fourier coefficients that heavily depend on the majority function, and after the first gradient step, coordinates that are on the parity have substantially larger gradient than those that aren't. The separation of input bits on the parity and those that aren't allows the analysis to establish that after the first iteration of gradient descent, the network has the capacity to represent the given parity, a representation theorem. Similarly, in our analysis we exploit the Fourier coefficients of hidden units, with the ReLU as a special case of a more general class of activation functions; see definition 4.7. In contrast, we show that on each iteration of gradient descent the gradient correlates with the parity, since hidden units have large Fourier coefficients on parities. Therefore, a representation theorem isn't needed. See Theorem 4.12.

## 3 PRELIMINARIES AND SETTING

Let $\mathcal{H}$ be the set of Boolean functions, $f : \{-1,1\}^n \longrightarrow \mathbb{R}$. Throughout the paper we will be approximating functions in $\mathcal{H}$ using neural networks with loss functions on the uniform hypercube, $\{-1,1\}^n$. An important Boolean function, the parity on $S \subset [n]$, is given by $\chi_S(x) = \prod_{i \in S} x_i$. Another important Boolean function is the majority function, defined when $n$ is odd: $Maj : \{-1,1\}^n \longrightarrow \{-1,1\}$. The majority function is 1 if there are more 1s than $-1$'s in a Boolean vector, and $-1$ otherwise. Below we present the following important fact from Boolean analysis that will be used throughout this work:

**Fact 3.1.** *Any Boolean function $f : \{-1,1\}^n \longrightarrow \mathbb{R}$ can be decomposed uniquely as follows:*

$$f(x) = \sum_{S \subset n} \widehat{f}(S) x_S(x)$$

*We call $\widehat{f}(S)$ the Fourier coefficient of $f$ on $S$. The set of Fourier coefficients of $f$ is called the Fourier spectrum of $f$. $\deg f$ is the size of the largest subset with a non-zero Fourier coefficient. See Theorem 1.1 O'Donnell (2021).*

We will be giving learning guarantees with the mean squared loss: $l(y, \hat{y}) = (y - \hat{y})^2$ and the hinge loss: $l(y, \hat{y}) = \max(0, 1 - y\hat{y})$. Given $g \in \mathcal{H}$ we will learn $g$ using gradient descent with a neural network $f_i$ where $f_i$ denotes the network at iteration, $i \geq 0$, of gradient descent. We initialize $f_0 = 0$. We will minimize the mean-squared loss: $\mathbb{E}_{x \sim \{-1,1\}^n}(g(x) - f_i(x))^2$ and the hinge loss: $\mathbb{E}_{x \sim \{-1,1\}^n} \max(0, 1 - g(x)f_i(x))$. Our neural networks will be of the form: $f(x) = \sum_{u \in W} a(u)\sigma(u \cdot x)$ where $W$ is the set of hidden weights, $a(u) \in \mathbb{R}$ is the output weight corresponding to the weight, $u$, and $\sigma : \mathbb{R} \longrightarrow \mathbb{R}$ is an activation function. We will define the operator, $\mathcal{J}_\sigma : \mathcal{H} \longrightarrow \mathcal{H}$ as follows, which we will refer to as $\mathcal{J}$ when the context is clear:

$$\mathcal{J}_\sigma(h)(u) = \mathbb{E}_{x \sim \{-1,1\}^n}[h(x)\sigma(u \cdot x)]$$

We will refer to each $\sigma(u \cdot .)$ as a unit and will often write $\sigma_u$ instead of $\sigma(u \cdot .)$. $\sigma_\mathbf{1}$ denotes the unit on $\mathbf{1} := \sum_i^n = 1^n e_i$. The concrete activation functions we will be using are a scaled ReLU and the sign function. We will also denote the indicator function as $\mathbf{1}(P)$, where $P$ is a truth value. If $P$ is true, then $\mathbf{1}(P) = 1$, otherwise 0. The weights of units will always be of the form $u \in \{-1,1\}^n$. We consider Linear Threshold Functions of the form $sign(u \cdot .)$ where $u \in \{-1,1\}^n$. The ReLU neurons we consider are of the form $\frac{1}{n}\max(0, u \cdot .)$ where $u \in \{-1,1\}^n$. However, our results are for more general classes of activation functions where ReLU neurons and Linear Threshold Functions are just special cases.

Facts 3.2 and 3.3 below make much use of the results of O'Donnell (2021) on Fourier coefficients of the majority function. Both ReLU activation functions and Linear Threshold functions have

spectrums that are highly dependent on the Fourier spectrum of the majority as shown in examples 4.9 and 4.10. Moreover, the Fourier coefficients of the majority function will be a central tool of the neural network training guarantees of theorems 4.2 and 4.4.

**Fact 3.2.** *Let $n$ be odd. Let $S, T \subset [n]$.*

*1.) If $|S| = |T|$ then $\widehat{Maj}(S) = \widehat{Maj}(T)$ (see Theorem 5.19 of O'Donnell (2021)).*

*2.) Fourier coefficients on even subsets of the majority function are $0$ (see Theorem 5.19 of O'Donnell (2021)).*

*3.) Suppose that $n \geq 2k^2$ where $k$ is odd. Then $W^k(Maj) = (\frac{2}{\pi})^{\frac{3}{2}} k^{-1/2}(1 \pm 1/k)$. Thus for any odd $k$-sized subset, $S$, $\widehat{Maj}(S)^2 = W^k(Maj)/\binom{n}{k} = n^{-O(k)}$ (See Corollary 5.23 of O'Donnell (2021))*

**Fact 3.3.** *Suppose $n, k$ are odd, $S \subset [n]$, and $n \geq 2k^2$. Consider $ReLU(u \cdot x) := \max(0, u \cdot x)$, and suppose $u \in \{-1, 1\}^n$. If $|S| = k + 1$, then the magnitude of the Fourier coefficients of $ReLU(u\cdot.)$ on $S$ is $\frac{1}{2}|\widehat{Maj}[k]|$. For simplicity, we will denote the Fourier coefficients of $ReLU(u\cdot.)$ as $\widehat{ReLU}(S)$. If $|S|$ is not an even and positive number, then*

$$\widehat{ReLU}(S) = 0 \text{ if } |S| > 1 \text{ is odd.}$$

$$|\widehat{ReLU}(\{i\})| = \frac{1}{2} \text{ for } i \in [n]$$

$$\widehat{ReLU}(\emptyset) = \frac{I[Maj]}{2} \text{ such that } \frac{\sqrt{n}}{\sqrt{2\pi}} \leq \widehat{ReLU}(\emptyset) \leq \frac{\sqrt{n}}{\sqrt{2\pi}} + O(n^{-\frac{1}{2}})$$

*Note that $I[Maj]$ is the total influence of the majority function. See definition 2.27 of O'Donnell (2021).*

*Proof.* First note that $ReLU(u \cdot x) = ReLU(\mathbf{1} \cdot x \circ u) = \sum_{S \subset [n]} \widehat{ReLU(\mathbf{1} \cdot .)}(S)\chi_S(u)\chi_S(x)$. Thus $|\widehat{ReLU(\mathbf{1} \cdot .)}(S)| = |\widehat{ReLU(u \cdot .)}(S)|$.

If $S = \emptyset$, then

$$|\widehat{ReLU(u \cdot .)}(\emptyset)| = \text{ (see Fact 1.12 of O'Donnell (2021)}$$

$$|\mathbb{E}_{x \sim \{-1,1\}^n} ReLU(\mathbf{1} \cdot x)| = |\mathbb{E}_{x \sim \{-1,1\}^n}(\mathbf{1} \cdot x)\mathbf{1}(\mathbf{1} \cdot x \geq 0)| =$$

$$|\mathbb{E}_{x \sim \{-1,1\}^n} \frac{(\mathbf{1} \cdot x)(1 + sign(\mathbf{1} \cdot x)}{2}| = |\mathbb{E}_{x \sim \{-1,1\}^n} \frac{(\mathbf{1} \cdot x)Maj(x)}{2}| =$$

$$|\frac{1}{2}\sum_{i=1}^n \mathbb{E}_{x \sim \{-1,1\}^n} \widehat{Maj}(i)x_i| = \frac{1}{2}\sum_{i=1}^n \widehat{Maj}(i) = \frac{1}{2}I[Maj]$$

Noting that $\frac{\sqrt{2n}}{\sqrt{\pi}} \leq I[Maj] \leq \frac{\sqrt{2n}}{\sqrt{\pi}} + O(n^{-\frac{1}{2}})$ by exercise 2.22e of O'Donnell (2021).

If $S = \{i\}$ then

$$\|\widehat{ReLU(u \cdot .)}(i)| = \frac{1}{2}|\mathbb{E}_x x_i \sum_j x_j(1 + Maj(x))| =$$

$$|\mathbb{E}_x x_i^2/2 + \mathbb{E}_x \sum_{j \neq i} x_i x_j Maj(x)/2| = \frac{1}{2}$$

since the majority is $0$ on even Fourier coefficients.

If $|S| > 1$ and odd then

$$|\widehat{ReLU(u \cdot .)}(S)| = |\mathbb{E}_x \chi_S(x)\frac{(\mathbf{1} \cdot x)(1 + Maj(x))}{2}| = |\mathbb{E}_x \chi_S(x)\frac{(\mathbf{1} \cdot x)Maj(x)}{2}| =$$

$$|\frac{1}{2}\sum_{i=1}^n \mathbb{E}_x \chi_S(x)x_i Maj(x)| = |\frac{1}{2}\sum_{i=1}^n \widehat{Maj}(T_i)| = 0$$

such that $T_i$ is $S - \{i\}$ if $i \in S$ and $S \bigcup \{i\}$, otherwise

noting that the last equality comes from Fact 3.2 #2.

If $|S| =: l > 1$ and even then

$$|\widehat{ReLU}(u \cdot .)(S)| = |\mathbb{E}_x\chi_S(x)\frac{(\mathbf{1} \cdot x)(1 + Maj(x))}{2}| = |\mathbb{E}_x\chi_S(x)\frac{(\mathbf{1} \cdot x)Maj(x)}{2}| =$$

$$|\frac{1}{2}\sum_{i=1}^{n}\mathbb{E}_x\chi_S(x)x_iMaj(x)| = \frac{1}{2}|\sum_{i\in S}\widehat{Maj}([l-1]) + \sum_{i\in[n]-S}\widehat{Maj}([l+1]))|$$

where the last equality above comes from Fact 3.2 # 1. By the proof of Lemma 2 in Barak et al. (2022), we have that

$$|\widehat{Maj}([l-1])| = \frac{n-l}{l-1}|\widehat{Maj}([l+1])|$$

Moreover, by Theorem 5.9 of O'Donnell (2021), $sign(\widehat{Maj}([l+1])) = -sign(\widehat{Maj}([l-1]))$. Thus

$$|\widehat{ReLU}(u \cdot .)(S)| = \frac{1}{2}|\sum_{i\in S}\widehat{Maj}([l-1]) - \sum_{i\in[n]-S}\frac{l-1}{n-l}\widehat{Maj}([l-1])(x)| =$$

$$\frac{1}{2}|l\widehat{Maj}([l-1]) - (n-l)\frac{l-1}{n-l}\widehat{Maj}([l-1])(x)| = \frac{1}{2}|\widehat{Maj}[l-1]|$$

$\square$

## 3.1 TRAINING PROCEDURES

To train our networks we draw a set, $W$, of $m$ weights independently from $\{-1,1\}^n$, draw a set of $m$ samples, $X$, independently from $\{-1,1\}^n$, and set $a_i(u) = 0$ for all $u \in W$.

### 3.1.1 MEAN SQUARED LOSS

To minimize the mean squared loss, we will update the linear output weights, $a(u)$, on each iteration of gradient descent by taking the derivative of the empirical mean squared loss:

$$\frac{1}{m}\sum_{x\in X}(g(x) - f_i(x))^2$$

Thus with a learning rate of $\frac{1}{2m}$ the update rule at iterate $i \geq 1$ of gradient descent for each $u \in W$ is

$$a_i(u) = a_{i-1}(u) - (-\frac{1}{2m}\frac{1}{m}\sum_{x\in X}2(g(x) - f_{i-1}(x))\sigma(u \cdot x)) = \quad (1)$$

$$a_{i-1}(u) + \frac{1}{m^2}\sum_{x\in X}(g(x) - f_{i-1}(x))\sigma(u \cdot x) \quad (2)$$

Thus we can write $f_i$ in terms of $f_{i-1}$ for $i \geq 1$ as follows:

$$f_i = \sum_{u\in W}a_i(u)\sigma_u = \sum_{u\in W}(a_{i-1}(u) + \frac{1}{m^2}\sum_{x\in X}(g(x) - f_{i-1}(x))\sigma(u \cdot x)\sigma_u =$$

$$f_{i-1} + \frac{1}{m^2}\sum_{u\in W}\sum_{x\in X}(g(x) - f_{i-1}(x))\sigma(u \cdot x)\sigma_u$$

For a finite set, $S \subset \mathbb{R}^n$, define the operator, $T_S : \mathcal{H} \to \mathcal{H}$, with respect to $\sigma$ as

$$T_S(h)(z) = \frac{1}{|S|}\sum_{x\in S}h(x)\sigma(x \cdot z)$$

From above, we've established the following proposition:

**Proposition 3.4.** *Using the update rule above with the mean squared loss, the $i \geq 1$ iteration of gradient descent yields*

$$f_i = f_{i-1} + T_W(T_X(g - f_{i-1}))$$

### 3.1.2 HINGE LOSS

Now, we establish the update rules for the hinge loss and obtain some results in a way similar to the above. With a learning rate of $\frac{1}{m}$, the update rule for gradient descent with the hinge loss at iteration $i \geq 1$ for any $u \in W$ is:

$$a_i(u) = a_{i-1}(u) + \frac{1}{m^2} \sum_{x \in X} \mathbf{1}(1 - g(x)f_{i-1}(x) \geq 0)g(x)\sigma_u(x) \tag{3}$$

Hence

$$f_i = f_{i-1} + \frac{1}{m} \sum_{u \in W} \frac{1}{m} \sum_{x \in X} \mathbf{1}(1 - f_{i-1}(x)g(x) \geq 0)g(x)\sigma(x \cdot u)\sigma_u$$

Thus we have the analogous results to Proposition 3.4 for the hinge loss, and get the following proposition:

**Proposition 3.5.** *Using the hinge loss and the update rule above, on iteration $i \geq 1$ of gradient descent, it follows that*

$$f_i = f_{i-1} + T_W(T_X(\mathbf{1}(1 - f_{i-1}g \geq 0)g))$$

## 4 RESULTS

Below we give agnostic upper bounds for learning functions on the hypercube via gradient descent on the mean squared loss. In the mean squared loss setting, we make use of similar techniques to the analysis in Vempala & Wilmes (2019). The two properties of their analysis that we draw analogies to are that the harmonic polynomials are an orthornomal basis for $L^2(\mathbb{S}^{n-1})$ and that the harmonic polynomials are eigenfunctions of Heck transforms. It is well known that the parities form an orthonormal basis for $L^2(\{-1,1\}^n)$; see Fact A.1. We draw the other analogy, in Proposition 4.1 below, by showing that the parites are eigenfunctions of the operator $\mathcal{J}_\sigma$, which depends on the activation function, $\sigma$.

**Proposition 4.1.** *Let $\sigma : \mathbb{R} \longrightarrow \mathbb{R}$ be an activation function, and define the operator $\mathcal{J}_\sigma : \mathcal{H} \longrightarrow \mathcal{H}$ by $\mathcal{J}_\sigma(g)(w) = \mathbb{E}_{x \sim \{-1,1\}^n} g(x)\sigma(x \cdot w)$. Then $\mathcal{J}_\sigma(g) = \sum_{S \subset [n]} \widehat{\sigma_\mathbf{1}}(S)\widehat{g}(S)\chi_S$. Thus the Fourier coefficients of $\mathcal{J}_\sigma(g)$ are: $\widehat{\mathcal{J}_\sigma(g)}(S) = \widehat{\sigma_\mathbf{1}}(S)\widehat{g}(S)$ where $S \subset [n]$.*

*Proof.* $\mathcal{J}_\sigma(g)(w) = \mathbb{E}_{x \sim \{-1,1\}^n} g(x)\sigma(x \cdot w) = \mathbb{E}_{x \sim \{-1,1\}^n}(\sum_{S \subset [n]} \widehat{g}(S)\chi_S(x))\sigma(w \cdot x) =$

$\sum_{S \subset [n]} \widehat{g}(S)\mathbb{E}_{x \sim \{-1,1\}}\chi_S(x)\sigma(\mathbf{1} \cdot w \circ x) = \sum_{S \subset [n]} \widehat{g}(S)\mathbb{E}_{x \sim \{-1,1\}}\chi_S(x)^2\widehat{\sigma_\mathbf{1}}(S)\chi_S(w) =$
$\sum_{S \subset [n]} \widehat{g}(S)\widehat{\sigma_\mathbf{1}}(S)\chi_S(w)$ □

We define $h_{even}^{\leq k}$ to be the best degree $k$ approximation in mean squared loss with even degree and degree 1 support for $h \in \mathcal{H}$. We define $h_{odd}^{\leq k}$ to be the best degree $k$ approximation in mean squared loss with odd degree support. Below we show that networks with Linear Threshold Function and ReLU units can agnostically learn Boolean functions with the mean squared loss.

**Theorem 4.2.** *Let $g \in \mathcal{H}$ and $\epsilon > 0$. Suppose that $n \geq 2(\deg g)^2$ and that $n$ is odd. Let $W$ and $X$ be sets of $m$ i.i.d. weights and $m$ i.i.d. samples drawn respectively from $\{-1,1\}^n$. Suppose $\|g - g_{odd}^{\leq k}\|_2 \leq \epsilon_{odd}$ and $\|g - g_{even}^{\leq k}\|_2 \leq \epsilon_{even}$ for some $\epsilon_{odd}$ and $\epsilon_{even}$. If $m = n^{O(k)}poly(\|g\|_2/\epsilon)$, then in time, $t = n^{O(k)} \log(\|g\|_2/\epsilon)$, a neural network updated with equation 1 with Linear Threshold Function units will converge in mean squared loss to $g$ with error at most $\epsilon + \epsilon_{odd}$ and a network with scaled ReLU units will converge in mean squared loss to $g$ with an error at most $\epsilon + \epsilon_{even}$, each with probability at least $1 - \frac{1}{m}$.*

*Proof.* This is a consequence of theorem 4.11 and the explanation of examples 4.9 and 4.10. □

With Theorem 4.2, it is a simple corollary to show that a class of 2-layer ReLU neural networks can be learned. Let $T \neq \emptyset \subset [n]$ and $f(T) : T \longrightarrow \{-1,1\}$ be a mapping. We define $u(f(T))$ to

be the vector in $\mathbb{R}^n$ such that $u(f(T))_i = f(T)(i)$, if $i \in T$, 0 otherwise. We say that $u(f(T))$ is supported on $T$.

Let $\mathcal{T}_r^B = \{T_s\}_{s=1}^r$ be an $r$-length sequence of odd sized subsets of $[n]$ such that, $|T_s| \le B$, for all $s \in [r]$. Define $M(\mathcal{T}_r^B) := \{f(T_s) : T_s \longrightarrow \{-1,1\}\}_{s=1}^r$. Let $a \in \mathbb{R}^r$ and let $\sigma$ be an activation function bounded by 1. We define $NN(M(\mathcal{T}_r^B), a, \sigma) := \sum_{s=0}^r a_s \sigma_{u(f(T_s))}$ to be the neural network on $\{-1,1\}^n$ with $r$ hidden $\sigma$ units of sparsity, $B$. We have that, $\|NN(M(\mathcal{T}_r^B), a, \sigma)\| \le \sum_{s=1}^r |a_s| \|\sigma_{u(f(T_s))}\| \le \|a\|_1$. Thus, we have the following corollary to Theorem 4.2. The theorem states that as target functions, neural networks with sparse hidden ReLU units, can be learned by neural networks trained by gradient descent on the mean squared loss.

**Theorem 4.3.** *Set $g := NN(M(\mathcal{T}_r^B), a, \sigma)$ where $\sigma$ is the scaled ReLU activation function. Let $\epsilon > 0$, $n \ge 2B^2$, and suppose that $n$ is odd. Let $W$ and $X$ be sets of $m$ i.i.d. weights and $m$ i.i.d. samples drawn respectively from $\{-1,1\}^n$. If $m = n^{O(B)} poly(\|a\|_1/\epsilon)$, then in time, $t = n^{O(B)} \log(\|a\|_1/\epsilon)$, a neural network updated with equation 1 with scaled ReLU units will converge in mean squared loss to $g$ with error at most $\epsilon$ with probability at least $1 - \frac{1}{m}$.*

*Proof.* The proof follows directly from Theorem 4.2. □

Below we give upper bounds for learning size $k$ parities with gradient descent on the hinge loss. We consider Linear Threshold Functions and ReLU units.

**Theorem 4.4.** *Let $S \subset [n]$ such that $|S| = k$ and $\epsilon > 0$. Suppose $n \ge 2k^2$ and that $n$ is odd. Let $W$ and $X$ be sets of $m$ i.i.d. weights and $m$ i.i.d. samples drawn respectively from $\{-1,1\}^n$. If $m = n^{O(k)} poly(1/\epsilon)$, then in time, $t = n^{O(k)} \log(1/\epsilon)$, a network updated with equation 3 with Linear Threshold Function units will have at most $\epsilon$ error on the hinge-loss with the parity on $S$ if $k$ is odd with probability at most $1 - \frac{1}{m}$. The same result holds for scaled ReLU networks, but when $k$ is even or odd when $k = 1$.*

*Proof.* This is a consequence of Theorem 4.12 and the explanation of examples 4.9 and 4.10. □

There has been empirical evidence that gradient descent learning with neural networks learns low-frequency Fourier components before high-frequency ones (Rahaman et al., 2019). On the theoretical side of "spectral bias", Cao et al. (2019) and Vempala & Wilmes (2019) have made strides in this direction. Below, definition 4.5 and the analysis of Theorem 4.6 are similar to definition 1.5 and Theorem 1.6 respectively in the latter.

Below we give guarantees that there is a spectral bias phenomenon in the Boolean hypercube setting and introduce some needed notation. Let $f \in \mathcal{H}$ and $\mathcal{S}$ be some collection of subsets of $[n]$. Denote $f^{\mathcal{S}} =: \sum_{S \in \mathcal{S}} \widehat{f}(S) \chi_S$ be the part of $f$ with Fourier coefficients in $\mathcal{S}$.

**Definition 4.5.** *Let $H_i = g - f_i$ denote the residual at iteration $i$ of training on the mean squared loss. Let $S, T \subset [n]$. We denote the change in residual at iteration $i$ by $\Delta_i = H_{i+1} - H_i$. Suppose $H_i^S, H_i^T \ne 0$. We define the rate of progress in $S$ relative to $T$ as $r_{i,S,T} > 0$ such that*

$$\frac{\|\Delta_i^S\|}{\|\Delta_i^T\|} = r_{i,S,T} \frac{\|H_i^S\|}{\|H_i^T\|}$$

*If $r_{i,S,T} > 1$, information on $S$ is learned more quickly than information on $T$ compared to what we would expect by the relative sizes of the residuals on $S$ and $T$. If $r_{i,S,T} < 1$, information on $T$ is learned more quickly.*

The next theorem shows that "lower frequencies" are learned more quickly than "higher frequencies". High frequencies, in this case, are large parities, and low frequencies are small parities.

**Theorem 4.6.** *Let $\epsilon > 0$ and suppose $g \in \mathcal{H}$ with $2(\deg g)^2 \le n$, $n$ odd. Assume that $g = g_{even}^{\le k}$. Let $S, T$ be subsets of $[n]$ in $g$'s support. Set $k := |S| < |T| =: l$ and assume that $\|H_i^S\|, \|H_i^T\| \ge \epsilon$. Consider the ReLU network with the training procedure of Theorem 4.2 with $m = n^{O(l)} poly(\|g\|_2/\epsilon)$ hidden weights and samples. Then with probability at least $1 - \frac{1}{m}$, the rate of progress of degree $S$ relative to $T$ is $r_{i,S,T} \ge n^{\Omega(l-k)}$*

*Proof.* See proof in appendix A.1. □

Below we introduce some terminology that will allow us to extend the results of Theorems 4.2 and 4.4 to more general activation functions.

**Definition 4.7.** *Let $\sigma$ be an activation function, and suppose that $\alpha > 0$. We say that $\sigma$ is an $(\mathcal{S}, \alpha)$ activation function if, for any $u \in \{-1, 1\}^n$, $|\widehat{\sigma_u}(S)| \geq \alpha$ for all $S \in \mathcal{S}$.*

**Remark 4.8.** *Definition 4.7 is equivalent to the following statement: there is some $u \in \{-1, 1\}^n$ such that $|\widehat{\sigma_u}(S)| \geq \alpha$ or any $S \in \mathcal{S}$.*

*Indeed, for any $w \in \{-1, 1\}^n$, $\sigma(w \cdot x) = \sigma(u \cdot (w \circ u \circ x)) = \sum_{S \subset [n]} \widehat{\sigma_u}(S) \chi_S(w \circ u) \chi_S(x)$. Hence, $\|\widehat{\sigma_w}(S) \chi_S\|_2 = \|\widehat{\sigma_u}(S) \chi_S(w \circ u) \chi_S\|_2 = \|\widehat{\sigma_u}(S) \chi_S\|_2 \geq \alpha$ for all $S \in \mathcal{S}$ as needed. The other direction is obvious.*

**Example 4.9.** *If $n$ is odd and $2k^2 \leq n$, the sign function is a $(\mathcal{S}, n^{-O(k)})$ activation function where $\mathcal{S}$ is the collection of odd subsets of $[n]$ of size at most $k$. The sign function is an $(\mathcal{S}, n^{-O(k)})$ activation function since for any $u \in \{-1, 1\}^n$, $|\widehat{sign(u \cdot .)}(S)| = |\widehat{Maj}(S)|$ for all $S \subset [n]$. Moreover, for all $S \in \mathcal{S}$, $|\widehat{sign(u, .)}(S)| = n^{-O(|S|)} \geq n^{-O(k)}$ (see Fact 3.2, #3).*

**Example 4.10.** *If $n$ is odd and $2k^2 \leq n$, the scaled ReLU function, $\frac{1}{n} \max(0, x)$, is a $(\mathcal{S}, n^{-O(k)})$ activation function where $\mathcal{S}$ is the collection of subsets of size one and all even subsets of $[n]$ of size at most $k$. The scaled ReLU function is an $(\mathcal{S}, n^{-O(k)})$ activation function since for any $u \in \{-1, 1\}^n$, $|\frac{1}{n} \widehat{\max(u \cdot ., 0)}(S)| = \frac{1}{2n} |\widehat{Maj}([\#S - 1])| \geq n^{-O(|S|)}$ for all $S \in \mathcal{S}$ such that $|S| > 1$. If $S \in \mathcal{S}$ such that $|S| = 1$ then $|\frac{1}{n} \widehat{\max(u \cdot ., 0)}(S)| = \frac{1}{2n}$. If $|S| = 0$ then $|\frac{1}{n} \widehat{\max(u \cdot ., 0)}(\emptyset)| = \frac{1}{\Theta(\sqrt{n})}$. See Fact 3.3 for results about the Fourier coefficients of ReLU units. We conclude that for all $S \in \mathcal{S}$, $|\frac{1}{n} \widehat{\max(u, ., 0)}(S)| = n^{-O(|S|)} \geq n^{-O(k)}$ (see Fact 3.2, #3).*

Below we extend the results of Theorem 4.2 to $(\mathcal{S}, \alpha)$ activation functions. In fact, Theorem 4.2 is a corollary of the result below by the discussions of Examples 4.9 and 4.10.

**Theorem 4.11.** *Let $g \in \mathcal{H}$ and $\epsilon > 0$. Let $W$ and $X$ be sets of $m$ i.i.d. weights and $m$ i.i.d. samples respectively drawn from $\{-1, 1\}^n$. Let $\sigma$ be an $(\mathcal{S}, \alpha)$ activation function where $\|\sigma\|_\infty \leq 1$. Suppose that $\|g - g^{\mathcal{S}}\|_2 \leq \epsilon_0$ for some $\epsilon_0 > 0$ and consider a network with $\sigma$ units. Then if $m = poly(1/\alpha) poly(\|g\|_2/\epsilon)$, in time $t = poly(1/\alpha) \log(\|g\|_2/\epsilon)$, the network updated with equation 1 will converge in mean squared loss with $g$ to error $\epsilon + \epsilon_0$ with probability at least $1 - \frac{1}{m}$.*

*Proof.* See appendix A.1. □

We now give the proof idea of Theorem 4.11. Suppose that instead of the update at iteration, $i$: $f_i = f_{i-1} + T_W T_X(H_{i-1})$, we instead used the update $f_i = f_{i-1} + \mathcal{J}^2(H_{i-1})$. Then we would have that $\|H_i - H_{i-1} - \mathcal{J}^2(H_{i-1})\|_2 = 0$. From Proposition A.3, it follows that

$$\|H_i\|_2^2 \leq \|H_{i-1} - \mathcal{J}^2(H_{i-1})\|_2^2 =$$
$$\|(1 - \mathcal{J}^2)(H_{i-1})\|_2^2 \leq \|H_{i-1}\|^2 - \alpha^2 \|H_{i-1}^{\mathcal{S}}\|_2^2$$

Thus we can descend as long as $\|H_{i-1}^{\mathcal{S}}\|_2$ is above our target error.

We describe how $\mathcal{J}^2(H_i)$ is approximated by $T_W T_X(H_i)$ in mean squared loss well enough, so we can descend in the same way as above. To get a good approximation with high-probability, we use concentration inequalities to get high probability bounds for $\mathcal{J}(\sigma_x)$ by $T_W(\sigma_x)$, $\mathcal{J}(\sigma_x)$ by $T_W(\sigma_x)$, and $\mathcal{J}(g)$ by $T_X(g)$ for all $x \in X$ and $w \in W$ holding simultaneously. The idea is that the mean squared loss, $\|T_W T_X(H_i) - \mathcal{J}^2(H_i)\|_2^2$, is bounded above by linear combinations of $\mathcal{J}(g)$ approximated by $T_X(g)$, and linear combinations of the worst mean squared loss approximation for $x \in X$ and $w \in W$: $\max_{x \in X} \|T_W(\sigma_x) - \mathcal{J}(\sigma_x)\|_2^2$ and $\max_{w \in W} \|T_X(\sigma_w) - \mathcal{J}(\sigma_w)\|_2^2$. Since these approximations are all good, to ensure that $T_W T_X(H_i)$ approximates $\mathcal{J}^2(H_i)$ well enough, we just have to ensure that the coefficients in the linear combinations are small enough for all iterations of descent. We can show that the coefficients are small enough, since they depend on the $a_i(u)$ and the magnitude of previous residuals.

Below we extend the results of Theorem 4.4 to more general settings. As in the mean squared loss case, Theorem 4.4 is a corollary of the result below by the discussions of Examples 4.9 and 4.10.

**Theorem 4.12.** *Let $S \subset [n]$ and $\epsilon > 0$. Let $W$ and $X$ be sets of $m$ i.i.d. weights and $m$ i.i.d. samples, respectively, drawn from $\{-1, 1\}^n$. Let $\sigma$ be an activation function such that $\|\sigma\|_\infty \leq 1$ and for all $u \in \{-1, 1\}^n$, $\sigma_u$'s Fourier coefficient on $S$ is $\alpha > 0$ in magnitude. If $m = poly(1/\alpha)poly(1/\epsilon)$, then in time $t = poly(1/\alpha)\log(1/\epsilon)$, the network with $\sigma$ units will approximate the parity on $S$ to an error of $\epsilon$ with the hinge loss with high probability.*

*Proof.* See appendix A.2. $\qquad\qquad\qquad\qquad\qquad\qquad\qquad\qquad\qquad\qquad\qquad\quad\square$

We describe the proof idea of Theorem 4.12. It is similar to the proof idea of Theorem 4.11. Suppose first that instead of updating our $f_i = f_{i-1} + T_W T_X (\mathbf{1}(1 - f_{i-1}g \geq 0)g)$, we updated the loss as $f_i = f_{i-1} + \mathcal{J}^2(g)$. Then if $\mathbb{E}_x \max(1 - f_i(x)g(x), 0) > 0$

$$\mathbb{E}_x \max(1 - f_i(x)g(x), 0) = \mathbb{E}_x \max(1 - g(x)(f_{i-1}(x) + \mathcal{J}^2(g)(x)), 0) =$$
$$\mathbb{E}_x \max(1 - g(x)f_{i-1}(x), 0) - \alpha^2$$

since $\mathcal{J}^2(g)g = \alpha^2$. Hence, we could descend by $\alpha^2$. Thus, if we can get a good approximation of $\mathcal{J}^2(g)$ by $T_W T_X (\mathbf{1}(1 - f_{i-1}g \geq 0)g)$ in mean squared loss, we can descend by a constant factor of $\alpha^2$. Indeed, $T_W T_X (\mathbf{1}(1 - f_{i-1}g \geq 0)g)$ approximates $\mathcal{J}^2(g)$ well-enough with high-probability to ensure the desired descent. To guarantee the approximation, as in Theorem 4.11, we get good approximations of $\mathcal{J}(\sigma_x)$ by $T_W(\sigma_x)$, $\mathcal{J}(\sigma_x)$ by $T_W(\sigma_x)$, and $\mathcal{J}(g)$ by $T_X(g)$ for all $x \in X$ and $w \in W$.

The following are the exact definitions of variables in Theorem 4.12, $m = O(\log(1/\delta)/\delta^2)$ such that $\delta = O(\alpha^4 \epsilon^2/t^3)$, and $t = O(\alpha^{-2}\log(1/\epsilon))$. To establish the approximation of $\mathcal{J}^2(g)$ by $T_W T_X (\mathbf{1}(1 - f_{i-1}g)g)$, with the above definitions, we present the following approximation lemma, the main ingredient in the proof of theorem 4.12:

**Lemma 4.13.** *(Lemma A.6 in the appendix) Suppose the assumptions of Theorem 4.12 hold. Let $E_k$ be the event $\{\|f_k - k\mathcal{J}^2(g)\| \leq O(k\delta)\}$ for $k \geq 0$. Note that there is some $s \leq t$ such that $s\alpha^2 < 1 \leq (s+1)\alpha^2$. Then for any integer $i$ such that $1 \leq i \leq s$ given that events $E_0, E_1, ..., E_{i-1}$ hold, then with probability at least $1 - O(\frac{\delta t^2}{\alpha^4})$ it follows that $\|f_i - i\mathcal{J}^2(g)\| \leq O(i\delta)$.*

Thus, with Lemma A.6 with events $E_0, E_1, ..., E_{i-1}$ by induction on the chain rule, we know with high-probability exactly what function $f_i$ is approximating with at most $\epsilon$ error, over all iterations, $i$, of gradient descent. Moreover, using Bernoulli's inequality, with probability at least: $(1 - O(\frac{\delta t^2}{\alpha^4}))^s \geq 1 - O(\frac{\delta s t^2}{\alpha^4})$ after $s$ iterations of gradient descent, the hinge loss error will be almost exactly $1 - \alpha^2 s$.

## 5    CONCLUSION AND FUTURE DIRECTIONS

This work gave polynomial time guarantees for neural network training on the uniform Boolean hypercube. In the mean squared loss setting, we showed that gradient descent can agnostically learn any function on the hypercube in time proportional to the smallest Fourier coefficient of the hidden unit on the shared support with the function. In the hinge loss setting, we showed that gradient descent can learn parities in time proportional to the hidden unit's Fourier coefficient on that parity.

In addition, even though the weights of hidden layers are not optimized, by SQ lower bounds on the parity and noting that gradient descent is an SQ algorithm, our analysis is essentially the best you can do.

Future directions would be to extend these techniques to get logarithmic time bounds in error for learning all Boolean functions, not just parties, on other losses such as the hinge loss. In this work it would be possible to not just agnostically learn degree $k$ polynomials in the mean-squared loss setting with gradient descent, but to learn all degree $k$ polynomials to arbitrary error, if our activation functions had support on every Fourier coefficient. Indeed, this work supports the latter if our activation function was the sum of a ReLU and a Linear Threshold function, although this is not a realistic assumption. We believe it would be an interesting future direction to get full Fourier support on activation functions with realistic model assumptions by potentially training both bias terms and the hidden weights in these units.

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

## A APPENDIX

The appendix includes results from Boolean analysis, and proofs of theorems 4.11 and 4.12 in sections A.1 and A.2 respectively.

**Fact A.1.** *For $f, g \in \mathcal{H}$, define the inner product:*

$$< f, g >= \mathbb{E}_{x \sim \{-1,1\}^n}[f(x)g(x)]$$

*The $2^n$ parity functions are an orthonormal basis for $\mathcal{H}$. See Theorem 1.5 in O'Donnell (2021).*

**Fact A.2.** *(Parseval's Theorem) let $f$ be a Boolean function. Then*

$$< f, f >= \|f\|_2^2 = \sum_{S \in [n]} \widehat{f}(S)^2$$

*. Moreover if the image of $f$ is $\{-1, 1\}$ then*

$$\sum_{S \in [n]} \widehat{f}(S)^2 = 1$$

*See page 25 of O'Donnell (2021).*

## A.1 MEAN SQUARED LOSS GUARANTEES

*Proof.* Proof of theorem 4.11. As in equations 5, 6, and 7 in Vempala & Wilmes (2019), throughout this proof we define the following parameters with constants $c_m, c_\delta$, and $c_t$:

$$m = c_m \|g\|_\infty \log(\|g\|_\infty/\delta)/\delta^2$$
$$\delta = c_\delta \alpha^4 \epsilon^2 / (\|g\|_2^2 t^2)$$
$$t = c_t \alpha^{-4} \log(\|g\|_2/\epsilon)$$

The settings of $m, \delta$, and $t$ above are the specific values of these parameters in the statement of Theorem 4.11. The proof of Proposition A.3 below is similar to Lemma 2.5 in Vempala & Wilmes (2019).

**Proposition A.3.** *Let $g \in \mathcal{H}$. Let $\sigma$ be an $(\mathcal{S}, \alpha)$ activation function bounded by* 1. *Then*

$$\|g - \mathcal{J}_\sigma^2(g)\|_2^2 \leq \|g\|_2^2 - \alpha^2 \|g^\mathcal{S}\|_2^2$$

*Proof.*

$$\|g - \mathcal{J}_\sigma^2(g)\|_2^2 = \mathbb{E}_{x \sim \{-1,1\}^n}(g(x) - \mathcal{J}_\sigma^2(g)(x))^2 =$$

$$\mathbb{E}_{x \sim \{-1,1\}^n} \left( \sum_{S \subset [n]} \widehat{g}(S)\chi_S(x) - \widehat{\sigma_{\mathbf{1}}}(S)^2 \widehat{g}(S)\chi_S(x) \right)^2 \text{ (by Proposition 4.1)} =$$

$$\sum_{S \subset [n]} (1 - \widehat{\sigma_{\mathbf{1}}}(S)^2)^2 \widehat{g}(S)^2 \text{ (by Parseval's Theorem)} \leq$$

$$\|g\|_2^2 - \sum_{S \subset [n]} \widehat{\sigma_{\mathbf{1}}}(S)^2 \widehat{g}(S)^2 \leq \|g\|_2^2 - \sum_{S \in \mathcal{S}} \widehat{\sigma_{\mathbf{1}}}(S)^2 \widehat{g}(S)^2 \leq$$

$$\|g\|_2^2 - \alpha^2 \|g^\mathcal{S}\|_2^2$$

where the last inequality follows since $\sigma$ is an $(\mathcal{S}, \alpha)$ activation function. $\square$

Lemma A.4 is the same as Lemma 3.6 in Vempala & Wilmes (2019), and below we check that it still holds in the Boolean setting.

**Lemma A.4.** *Let $\sigma$ be bounded by* 1. *Then with probability $1 - m$ over the choice of $X$ and $W$, the following statements are all true:*

$$\text{1.) } \|T_X(g) - \mathcal{J}(g)\|_2 \leq \delta$$
$$\text{2.) for all } w \in W, \text{ we have } \|T_X(\sigma_w) - \mathcal{J}(\sigma_w)\|_2 \leq \delta$$
$$\text{3.) for all } x \in X, \text{ we have } \|T_W(\sigma_x) - \mathcal{J}(\sigma_x)\|_2 \leq \delta$$
$$\text{4.) for all } x \neq y \in X, \text{ we have } |T_W(\sigma_x)(y) - \mathcal{J}(\sigma_x)(y)| \leq \delta/2$$

*Proof.* The proof of this lemma depends on Lemma 3.5 of Vempala & Wilmes (2019), and except for Lemma 3.5, the proof doesn't have any dependence on the uniform sphere. It follows that we just need to check that the parts of the proof in Lemma 3.5 of Vempala & Wilmes (2019) that use data from the sphere still hold in the Boolean setting. We make this check below.

Fix $u \in \{-1,1\}^n$. Let $f$ be a Boolean function and draw $x_1, .., x_l$ i.i.d. from $\{-1,1\}^n$. Then $\zeta_1, ..., \zeta_l$ are i.i.d. where $\zeta_i = f(x_i)\sigma(u \cdot x_i) - \mathcal{J}(f)(u)$ for each $i$. Note that

$$\mathbb{E}_x[\zeta_i] = \mathbb{E}_x[f(x_i)\sigma(u \cdot x_i) - \mathcal{J}(f)(u)] =$$
$$\mathcal{J}(f)(u) - \mathcal{J}(f)(u) = 0$$

At this point, to have an identical proof of Lemma 3.5 of Vempala & Wilmes (2019), we just need to show that $\|\mathcal{J}(G)\|_\infty \leq \|\sigma\|_2 \|G\|_2$ for all $G \in \mathcal{H}$, $Var(\zeta_i) \leq \|f\|_2^2$, and $|\zeta_i| \leq 2\|f\|_\infty$. Indeed, $\|\mathcal{J}(G)\|_\infty = |\mathbb{E}_x \sigma(u^* \cdot x)G(x)| \leq \|\sigma_u\|_2 \|G\|_2$ by Cauchy Schwartz, $Var(\zeta_i) =$

$$\mathbb{E}_x(\zeta_i)^2 = \mathbb{E}_x(f(x_i)\sigma(u \cdot x_i) - \mathcal{J}(f)(u))^2 = Var(f(x_i)\sigma_u(x_i)) =$$
$$\mathbb{E}_x f(x_i)^2 \sigma_u(x_i)^2 \leq \mathbb{E}_x f(x_i)^2 = \|f\|_2^2.$$

In addition,
$$|\zeta_i| \leq \|f\sigma_u - \mathcal{J}(f)(u)\|_\infty \leq \|f\sigma_u\|_\infty \leq \|f\sigma_u\|_\infty + |\mathcal{J}(f)(u)| \leq \|f\|_\infty + \|\sigma_u\|_2\|f\|_2 \leq$$
$$2\|f\|_\infty.$$
$\square$

From now on, we essentially follow the proof of Theorem 1.3 of Vempala & Wilmes (2019) verbatim. We can prove Lemma 3.7 of Vempala & Wilmes (2019) with Lemmas A.4 and 3.4. Lemma 3.8 of Vempala & Wilmes (2019) follows from $\|\mathcal{J}(f)\|_\infty \leq \|\sigma\|_2\|f\|_2$, which we showed above, Proposition 3.4, and Lemma 3.7 of Vempala & Wilmes (2019). The proof of Lemma 3.3 of Vempala & Wilmes (2019) can be proven with Proposition A.3 and Lemma 3.8 of Vempala & Wilmes (2019). The proof of Theorem 1.3 in Vempala & Wilmes (2019) now follows from Proposition A.3 and Lemma 3.3 of Vempala & Wilmes (2019). $\square$

*Proof.* Proof of Theorem 4.6. The proof is similar to the proof of Theorem 1.6 in Vempala & Wilmes (2019) and since the conditions of Theorem 4.2 are satisfied, we can use Lemma 3.3 from their work, which states:
$$\|\Delta H_i - \mathcal{J}^2(H_i)\|_2 = O(\delta\|g\|_2 t^2)$$
We set $\sigma := \frac{1}{n}\max(0, x)$, the scaled ReLU. Denote the absolute value of the Fourier coefficient of the scaled ReLU on $V$, where $|V| = s$, by $\zeta_s := |\widehat{\sigma_1}(V)|$. By Fact 3.3, $\zeta_0 = \Theta(1/\sqrt{n})$, $\zeta_1 = \frac{1}{2n}$, and $\zeta_s = \frac{1}{2n}|\widehat{Maj}([s-1])|$ if $s > 1$ and $s$ is even.

Recall in the definition of $\delta$, $\alpha > 0$, where $\alpha \leq \zeta_k, \zeta_l$, noting that $|S| = k$ and $|T| = l$, and $T$ and $S$ are in $g$'s support.

We're now ready to carry out our analysis, and have that:
$$r := r_{i,S,T} = \frac{\|\Delta_i^S\|\|H_i^T\|}{\|\Delta_i^T\|\|H_i^S\|} \geq \frac{(\|\mathcal{J}^2 H_i^S\| - O(\delta t^2\|g\|))\|H_i^T\|}{(\|\mathcal{J}^2 H_i^T\| + O(\delta t^2\|g\|))\|H_i^S\|} =$$
$$\frac{(\zeta_k^2\|H_i^S\| - O(\delta t^2\|g\|))\|H_i^T\|}{(\zeta_l^2\|H_i^T\| + O(\delta t^2\|g\|))\|H_i^S\|}$$
First note that from the proof of Lemma 3.3 in Vempala & Wilmes (2019) that $\|H_i\| \leq \|g\|$. Recalling the definition of $\delta$, it follows that
$$(\zeta_l^2\|H_i^T\| + O(\delta t^2\|g\|))\|H_i^S\| \leq (\zeta_l^2\|H_i^T\| + \alpha^4\epsilon^2)\|H_i^S\| \leq$$
$$2\zeta_l^2\|H_i^S\|\|H_i^T\|$$
since $\zeta_l^2 \geq \alpha^4$, and $\epsilon^2 \leq \|H_i^T\|$ by assumption. Moreover
$$O(\delta t^2\|g\|\|H_i^T\|) \leq O(\delta t^2\|g\|^2) \leq \epsilon^2\alpha^4 \leq \frac{1}{2}\|H_i^S\|\zeta_k^2$$
Putting everything together,
$$r \geq \frac{1/2\zeta_k^2\|H_i^S\|\|H_i^T\|}{2\zeta_l^2\|H_i^T\|\|H_i^S\|} = \frac{\zeta_k^2}{4\zeta_l^2} = n^{\Omega(l-k)}$$
We show the final equality above. Recall from Fact 3.2 #3 that if $s > 1$ then $\widehat{Maj}([s-1])^2 = n^{-O(s)}$. Moreover, if $S$ or $T$ are of size bigger than 1, then the set must be even since $S, T$ are in $g$'s support.

If $k = 0$, then $\zeta_k^2 = \frac{1}{\Theta(n)}$ and $\zeta_l^{-2} = 4n^2$, if $l=1$. If $l > 1$ then $\zeta_l^{-2} = 1/n^{-O(l)} = n^{\Omega(l)}$. Either way, the claim holds.

If $k = 1$ then $\zeta_k^2 = \frac{1}{4n^2}$, and $\zeta_l^{-2} = n^{\Omega(l)}$ since $l$ must be even and bigger than 1. Thus, the claim holds.

If $k > 1$, both $S$ and $T$ must be of even size. Hence,
$$\frac{\zeta_k^2}{4\zeta_l^2} = \frac{\widehat{Maj}([k-1])^2}{\widehat{Maj}([l-1])^2} = n^{\Omega(l-k)}$$

$\square$

A.2    HINGE LOSS GUARANTEES

*Proof.* We prove Theorem 4.12 in this section. Throughout the discussion, $g$ is a parity, and the magnitude of $\sigma_u$'s Fourier coefficient on $g$ is $\alpha > 0$ for all $u \in \{-1, 1\}^n$. Throughout the proof, we will use the definitions of the following parameters with constants $c_m, c_\delta$, and $c_t$:

$$m = c_m \log(1/\delta)/\delta^2$$
$$\delta = c_\delta \alpha^4 \epsilon^2/t^3$$
$$t = c_t \alpha^{-2} \log(1/\epsilon)$$

The statement of Theorem 4.12 with the above values of $m, \delta$, and $t$ will be proven.

With the definitions of $m, \delta$, and $t$ above, with probability at least $1 - 1/m$, the statements in Lemma A.4 are all true. Indeed, in the proof Lemma 3.6 in Vempala & Wilmes (2019), Lemma A.4 in this work, their only requirement on $m$ is that $m \geq c_m \|g\|_\infty \log(\|g\|_\infty/\delta)/\delta^2$, and $\delta$ is arbitrary. Since $g$ is the parity, $\|g\|_\infty = 1$, and our $m$ above satisfies the requirements of Lemma A.4.

Lemma A.5, below, is almost identical to Lemma 3.7 of Vempala & Wilmes (2019), but uses the gradient of the hinge loss instead of the gradient of the mean squared loss. We include a proof below, and note that from Proposition 3.5, $f_i - f_{i-1} = T_W T_X(\mathbf{1}(1 - gf_{i-1} \geq 0)g)$.

**Lemma A.5.** *Suppose that conditions 1-3 of Lemma A.4 hold. Then for all $i \geq 1$ we have that*

$$\|(T_W - \mathcal{J})T_X(\mathbf{1}(1 - gf_{i-1} \geq 0)g)) \leq \delta$$

*Proof.*

$$\|(T_W - \mathcal{J})T_X(\mathbf{1}(1 - gf_{i-1} \geq 0)g))\| =$$
$$\|(T_W - \mathcal{J})(\frac{1}{m}\sum_{x \in X}\mathbf{1}(1 - g(x)f_{i-1}(x) \geq 0)g(x))\sigma_x\|_2 \leq$$
$$\frac{1}{m}\sum_{x \in X}\|(T_W - J)(\sigma_x)\|_2 \leq \delta$$

Where the last inequality comes from Lemma A.4 # 3.    □

The next lemma is the main technical ingredient for the proof of Theorem 4.12.

**Lemma A.6.** *Suppose that conditions 1-3 of Lemma A.4 hold. Let $E_k$ be the event $\{\|f_k - k\mathcal{J}^2(g)\| \leq O(k\delta)\}$ for $k \geq 0$. Note that there is some $s \leq t$ such that $s\alpha^2 < 1 \leq (s+1)\alpha^2$. Then for any integer $i$ such that $1 \leq i \leq s$ given that event $E_0, ..., E_{i-1}$ holds then with probability at least $1 - O(\frac{\delta t^2}{\alpha^4})$ it follows that*

$$\|f_i - i\mathcal{J}^2(g)\| \leq O(i\delta)$$

*Proof.* Suppose $i = 1$. The event $E_0$ holds trivially. Then $f_1 = T_W T_X(\mathbf{1}(1 - f_0 g \geq 0)g) = T_W T_X(g)$. By Lemma A.4 1.) and Lemma A.5, it follows that

$$\|f_1 - \mathcal{J}^2(g)\| \leq \|T_W T_X(g) - \mathcal{J}T_X(g)\| + \|\mathcal{J}T_X(g) - \mathcal{J}^2(g)\| \leq$$
$$\delta + \|T_X(g) - \mathcal{J}(g)\| \leq 2\delta$$

Noting that for any Boolean function $h$, $\|\mathcal{J}(h)\| \leq \|h\|$ since $\sigma$ is bounded by 1.

Now assume that $i > 1$. Since the event $E_{i-1}$ holds it follows that

$$\|f_i - i\mathcal{J}\| \leq \|f_{i-1} - (i-1)\mathcal{J}(g)\| +$$
$$\|T_W T_X(\mathbf{1}(1 - f_{i-1}g \geq 0)g) - \mathcal{J}^2(g)\| \leq O((i-1)\delta) +$$
$$\|T_W T_X(\mathbf{1}(1 - f_{i-1}g \geq 0)g) - \mathcal{J}^2(g)\|$$

We now bound $\|T_W T_X(\mathbf{1}(1 - f_{i-1}g \geq 0)) - \mathcal{J}^2(g)\|$ and have that

$$\|T_W T_X(\mathbf{1}(1 - f_{i-1}g \geq 0)g) - \mathcal{J}^2(g)\| \leq$$
$$\|T_W T_X(\mathbf{1}(1 - f_{i-1}g \geq 0)) - \mathcal{J}^2(\mathbf{1}(1 - f_{i-1}g \geq 0)g)\|$$
$$+\|\mathcal{J}^2(\mathbf{1}(1 - f_{i-1}g \geq 0)g) - \mathcal{J}^2(g)\|$$

We will now bound $\|\mathcal{J}^2(\mathbf{1}(1 - f_{i-1}g \geq 0)g) - \mathcal{J}^2(g)\|$. First note that:

$$\|\mathbf{1}(1 - f_{i-1}g \geq 0)g - g\|^2 = \|1 - \mathbf{1}(1 - f_{i-1}g \geq 0)\|^2 =$$
$$\Pr[\mathbf{1}(1 - f_{i-1}g \geq 0) = 0] = \Pr[f_{i-1}g > 1] \leq$$
$$\Pr[f_{i-1}g \geq 1] \leq \Pr[|f_{i-1} - (i-1)\mathcal{J}^2(g)| \geq \alpha^2]$$

The last inequality follows since if $f_{i-1}g \geq 1$, then

$(i-1)\mathcal{J}^2(g)g = (i-1)\alpha^2 g^2 = (i-1)\alpha^2 < (s-1)\alpha^2 < 1 - \alpha^2$, and it follows that

$f_{i-1}g - \alpha^2 \geq 1 - \alpha^2 > (i-1)\mathcal{J}^2(g)g$. Thus $f_{i-1}g - (i-1)\mathcal{J}^2(g)g \geq \alpha^2$ as needed.
By Markov's inequality and assumption, we have that

$$\Pr[|f_{i-1} - (i-1)\mathcal{J}^2(g)| \geq \alpha^2] \leq \|f_{i-1} - (i-1)\mathcal{J}^2(g)\|^2/\alpha^4 \leq$$
$$O((i-1)^2\delta^2/\alpha^4) = O(\delta)$$

by the definition of $\delta$.

Since $\|\mathcal{J}^2(\mathbf{1}(1 - f_{i-1}g \geq 0)g) - \mathcal{J}^2(g)\| \leq \|\mathcal{J}^2(\mathbf{1}(1 - f_{i-1}g \geq 0)g) - \mathcal{J}^2(g)\|_\infty$,

we will upper bound $\|\mathcal{J}^2(\mathbf{1}(1 - f_{i-1}g \geq 0)g) - \mathcal{J}^2(g)\|_\infty$. Let $u$ be an element of

the hypercube. Then

$$|\mathcal{J}^2(\mathbf{1}(1 - f_{i-1}g \geq 0)g - g)(u)| = |\mathbb{E}_v h(v)\sigma_u(v)| \leq \mathbb{E}_v|h(v)| =$$
$$\mathbb{E}_v|\mathbb{E}_x \mathbf{1}(1 - f_{i-1}(x)g(x) \geq 0)g(x) - g(x))\sigma_v(x)| \leq$$
$$\mathbb{E}_v \mathbb{E}_x|\mathbf{1}(1 - f_{i-1}(x)g(x) \geq 0)g(x) - g(x))\sigma_v(x)| =$$
$$\mathbb{E}_v \mathbb{E}_x|\mathbf{1}(1 - f_{i-1}(x)g(x) \geq 0) - 1| \cdot |g(x)\sigma_v(x)| \leq$$
$$\mathbb{E}_x|\mathbf{1}(1 - f_{i-1}(x)g(x) \geq 0) - 1| = \Pr[f_{i-1}(x)g(x) > 1] \leq O(\delta)$$

Hence $\|\mathcal{J}^2(\mathbf{1}(1 - f_{i-1}g \geq 0)g) - \mathcal{J}^2(g)\| \leq O(\delta)$ as needed.

We now bound $\|T_W T_X(\mathbf{1}(1 - f_{i-1}g \geq 0)g) - \mathcal{J}^2(\mathbf{1}(1 - f_{i-1}g \geq 0)g)\|$. We have that

$$\|T_W T_X(\mathbf{1}(1 - f_{i-1}g \geq 0)g) - \mathcal{J}^2(\mathbf{1}(1 - f_{i-1}g \geq 0)g)\| \leq$$
$$\|\|T_W T_X(\mathbf{1}(1 - f_{i-1}g \geq 0)g) - \mathcal{J}T_X(\mathbf{1}(1 - f_{i-1}g \geq 0)g)\|+$$
$$\|\mathcal{J}T_X(\mathbf{1}(1 - f_{i-1}g \geq 0)g) - \mathcal{J}^2(\mathbf{1}(1 - f_{i-1}g \geq 0)g\| \leq$$
$$\delta + \|\mathcal{J}T_X(\mathbf{1}(1 - f_{i-1}g \geq 0)g) - \mathcal{J}^2(\mathbf{1}(1 - f_{i-1}g \geq 0)g)\|$$

by Lemma A.5.

We now bound $\|\mathcal{J}T_X(\mathbf{1}(1 - f_{i-1}g \geq 0)g) - \mathcal{J}^2(\mathbf{1}(1 - f_{i-1}g \geq 0)g)\|$. Using the results from
Lemma A.4 1.) it follows that:

$$\|\mathcal{J}T_X(\mathbf{1}(1 - f_{i-1}g \geq 0)g) - \mathcal{J}^2(\mathbf{1}(1 - f_{i-1}g \geq 0)g)\| \leq$$
$$\|T_X(\mathbf{1}(1 - f_{i-1}g \geq 0)g) - \mathcal{J}(\mathbf{1}(1 - f_{i-1}g \geq 0)g)\| \leq$$
$$\|T_X(\mathbf{1}(1 - f_{i-1}g \geq 0)g) - T_X(g)\| + \|T_X(g) - \mathcal{J}(\mathbf{1}(1 - f_{i-1}g \geq 0)g)\| \leq$$
$$\|T_X(\mathbf{1}(1 - f_{i-1}g \geq 0)g) - T_X(g)\| + \|\mathcal{J}(g) - \mathcal{J}(\mathbf{1}(1 - f_{i-1}g \geq 0)g)\| + \delta \leq$$
$$\|T_X(\mathbf{1}(1 - f_{i-1}g \geq 0)g) - T_X(g)\| + \|\mathcal{J}(g) - \mathcal{J}(\mathbf{1}(1 - f_{i-1}g \geq 0)g)\| + \delta$$

From earlier we have:

$$|\mathcal{J}(g) - \mathcal{J}(\mathbf{1}(1 - f_{i-1}g \geq 0)g)(u)| = |\mathbb{E}_x(\mathbf{1}(1 - f_{i-1}g(x) \geq 0) - 1)g(x)\sigma_u(x)| \leq$$
$$\mathbb{E}_x|(\mathbf{1}(1 - f_{i-1}g(x) \geq 0) - 1)| =$$
$$\Pr[f_{i-1}(x)g(x) > 1] \leq O(\delta)$$

Thus,

$$\|\mathcal{J}T_X(\mathbf{1}(1 - f_{i-1}g \geq 0)g) - \mathcal{J}^2(\mathbf{1}(1 - f_{i-1}g \geq 0)g)\| \leq$$
$$\|T_X(\mathbf{1}(1 - f_{i-1}g \geq 0)g) - T_X(g)\| + O(\delta)$$

Note that for a Boolean function, $h$, $\|T_X(h)\| = \|\frac{1}{m}\sum_{x \in X} h(x)\sigma_x\| \leq \frac{1}{m}\sum_{x \in X} |h(x)|$ since $\|\sigma_x\| \leq 1$. Thus it follows that:

$$\|T_X(\mathbf{1}(1 - f_{i-1}g \geq 0)g) - T_X(g)\| \leq \frac{1}{m}\sum_{x \in X} |(\mathbf{1}(1 - f_{i-1}(x)g(x) \geq 0) - 1)g(x)| =$$
$$\frac{1}{m}\sum_{x \in X} 1 - \mathbf{1}(1 - f_{i-1}(x)g(x) \geq 0)$$

Now we calculate the probabilistic conclusion of the lemma. Using Markov's inequality we get:

$$\Pr[\frac{1}{m}\sum_{x \in X} 1 - \mathbf{1}(1 - f_{i-1}(x)g(x) \geq 0) \geq \delta] \leq \frac{1 - \mathbb{E}\mathbf{1}(1 - f_{i-1}(x)g(x) \geq 0)}{\delta}$$

We upper bound $\mathbb{E}\mathbf{1}(1 - f_{i-1}(x)g(x) \geq 0) = 1 - \Pr[1 - f_{i-1}(x)g(x) < 0]$. From earlier in the proof, we have that

$$\Pr[1 - f_{i-1}(x)g(x) < 0] \leq \Pr[f_{i-1}(x)g(x) \geq 1] \leq \Pr[|f_{i-1} - (i-1)\mathcal{J}^2(g)| \geq \alpha^2] =$$
$$O((i-1)^2\delta^2/\alpha^4)$$

Hence,

$$\frac{1 - \mathbb{E}\mathbf{1}(1 - f_{i-1}(x)g(x) \geq 0)}{\delta} = \frac{\Pr[1 - f_{i-1}(x)g(x) < 0]}{\delta} = O(i-1)^2\delta/\alpha^4$$

Thus the probability that $\|T_X(\mathbf{1}(1 - f_{i-1}g \geq 0)g) - T_X(g)\| < \delta$ is at least $1 - O(i-1)^2\delta/\alpha^4$. Hence $\|\mathcal{J}T_X(\mathbf{1}(1 - f_{i-1}g \geq 0)g) - \mathcal{J}^2(\mathbf{1}(1 - f_{i-1}g\| \leq O(\delta)$ with probability at least $1 - O(i-1)^2\delta/\alpha^4$. Thus

$$\|T_W T_X(\mathbf{1}(1 - f_{i-1}g \geq 0)g) - \mathcal{J}^2(\mathbf{1}(1 - f_{i-1}g \geq 0)g)\| \leq O(\delta)$$

with probability at least $1 - O((i-1)^2\delta/\alpha^4)$.

Hence

$$\|T_W T_X(\mathbf{1}(1 - f_{i-1}g \geq 0)g) - \mathcal{J}^2(g)| \leq$$
$$\|T_W T_X(\mathbf{1}(1 - f_{i-1}g \geq 0)) - \mathcal{J}^2(\mathbf{1}(1 - f_{i-1}g \geq 0)g)\|+$$
$$\|\mathcal{J}^2(\mathbf{1}(1 - f_{i-1}g \geq 0)g) - \mathcal{J}^2(g)\| = O(\delta)$$

and putting everything together we get that,

$$\|f_i - i\mathcal{J}^2 J(g)\| \leq \|f_{i-1} - (i-1)\mathcal{J}^2(g)\| + \|T_W T_X(\mathbf{1}(1 - f_{i-1}g \geq 0)g) - \mathcal{J}^2(g)\| =$$
$$O((i-1)\delta) + O(\delta) = O(i\delta)$$

with probability at least $1 - O((i-1)^2\delta/\alpha^4)$ completing the proof. $\square$

Below is the statement of Theorem 4.12 and the remaining part of the proof with the above lemmas of this section at hand.

**Theorem A.7.** *Let $\sigma$ be an activation function such that the magnitude of it's Fourier coefficient on $g = \chi_S$ is $\alpha > 0$ and $\|\sigma\|_\infty \leq 1$. Assume that $m = O(\log(1/\delta)/\delta^2)$ where $\delta = O(\alpha^4\epsilon^2/t^3)$, and $t = O(\alpha^{-2}\log(1/\epsilon))$. Then with probability at least $(1 - 1/m)(1 - O(\delta t^3/\alpha^4))(1 - O(t^2\delta/\epsilon^2))$ a network with $m$ randomly initialized $\sigma$ neurons will converge to $\epsilon$ error on the hinge loss in at most $t$ iterations of gradient descent using the same set of $m$ samples on each iteration.*

*Proof.* There is some $s \leq t$ such that $s\alpha^2 < 1 \leq (s+1)\alpha^2$. In either case, with probability at least $1 - \frac{1}{m}$, the conditions of A.4 hold. There are two cases: $1 - \epsilon/2 < s\alpha^2 < 1$ and $s\alpha^2 \leq 1 - \epsilon/2$. We first explore the case, $1 - \epsilon/2 < s\alpha^2 < 1$. Then from Lemma A.6, if $E_0, E_1, .., E_{s-1}$ are true, it follows that

$$\|f_s - s\mathcal{J}^2(g)\| \leq O(s\delta)$$

We now upper bound the hinge loss on $f_s$ by $\epsilon$. By Jensen's inequality, that $\max$ is 1-Lipschitz, and by Lyapunov's inequality we have that:

$$|\mathbb{E}\max(0, 1 - f_s(x)g(x))) - \mathbb{E}\max(0, 1 - s\mathcal{J}^2(x)g(x))| =$$
$$|\mathbb{E}[\max(0, 1 - f_s(x)g(x)) - \max(0, 1 - s\mathcal{J}^2(x)g(x))]| \leq$$
$$\mathbb{E}|\max(0, 1 - f_s(x)g(x)) - \max(0, 1 - s\mathcal{J}^2(x)g(x))| \leq \mathbb{E}|f_s(x)g(x) - s\mathcal{J}^2(x)g(x)| \leq$$
$$\|f_s - s\mathcal{J}^2(g)\|$$

Thus

$$\mathbb{E}\max(0, 1 - f_s(x)g(x)) \leq O(s\delta) + \mathbb{E}\max(0, 1 - s\mathcal{J}^2(x)g(x)) < \epsilon/2 + \epsilon/2 = \epsilon$$

choosing the constant in the definition of $\delta$ suitably and noting that $1 - s\mathcal{J}^2(x)g(x) = 1 - s\alpha^2 < 1 - (1 - \epsilon/2) = \epsilon/2$. Now, by Lemma A.6 with events $E_0, .., E_{s-1}$, by induction on the chain rule, and Bernoulli's inequality, with probability at least

$$(1 - 1/m)(1 - O(\delta t^2/\alpha^4))^t \geq (1 - 1/m)(1 - O(\delta t^3/\alpha^4))$$

that after at most $t$ iterations of gradient descent, the error on the hinge loss will be less than $\epsilon$. The case when $1 - \epsilon/2 < s\alpha^2 < 1$ is now complete.

We now move onto the second case, $s\alpha^2 \leq 1 - \epsilon/2$. Using the same argument as above, we have that with probability at least $(1 - 1/m)(1 - O(\delta t^3/\alpha^4))$ that

$$\|f_s - s\mathcal{J}^2(g)\| \leq O(s\delta)$$

Thus, we have that,

$$\|f_{s+1} - (s+1)\mathcal{J}^2(g)\| \leq O(s\delta) + \|T_W T_X(\mathbf{1}(1 - f_s g \geq 0)g) - \mathcal{J}^2(g)\|$$

Everything for this case is the same as in Lemma A.6, except that we must upper bound

$\|T_W T_X(\mathbf{1}(1 - f_s g \geq 0)g) - \mathcal{J}^2(g)\|$ by $\epsilon/2$ by bounding

$\|\mathcal{J}^2(\mathbf{1}(1 - f_s g \geq 0)g) - \mathcal{J}^2(g)\| \leq \Pr[f_s g \geq 1] \leq \Pr[|f_s - s\mathcal{J}^2(g)| \geq \epsilon/2]$

instead of $\Pr[|f_{i-1} - (i-1)\mathcal{J}^2(g)| \geq \alpha^2]$ as in the proof of Lemma A.6 and noting that if $f_s g \geq 1$, since $s\mathcal{J}^2(g)g = s\alpha^2 \leq 1 - \epsilon/2$, it follows that $f_s g - s\mathcal{J}^2(g)g > \epsilon/2$. Similarly, we upper bound

$$\Pr[1 - f_s(x)g(x) < 0] = O(s^2\delta^2 4/\epsilon^2)$$

instead of the upper bound $O((i-1)^2\delta/\alpha^4)$ in the proof of Lemma A.6. Putting everything together, we get with probability at least $1 - O(s^2\delta/\epsilon^2)$ that

$$\|T_W T_X(\mathbf{1}(1 - f_{i-1}g \geq 0)g - \mathcal{J}^2(g)\| \leq O(\delta)$$

Finally, putting everything together and from similar arguments in the first case, since

$$\|f_{s+1} - (s+1)\mathcal{J}^2(g) \leq O((s+1)\delta)$$

and since $\mathbb{E}\max(0, 1 - (s+1)\mathcal{J}^2(g)(x)g(x)) = 0$, it follows that

$$\mathbb{E}\max(0, 1 - f_{s+1}(x)g(x)) < O((s+1)\delta) + 0 = O((s+1)\delta) < \epsilon$$

for a suitably small choice of $\delta$ with probability at least

$$(1 - 1/m)(1 - O(\delta t^3/\alpha^4))(1 - O(t^2\delta/\epsilon^2))$$

completing the proof. □

□

