# OpenReview forum: "Learning Boolean functions with neural networks"
_ICLR.cc/2024/Conference — Submitted to ICLR 2024_

### Official Review · Reviewer_LdZ7 · 2023-10-30

**Soundness:** 2 fair
**Presentation:** 1 poor
**Contribution:** 2 fair
**Rating:** 3
**Confidence:** 3

**Summary:**

This paper examines the learning of Boolean functions using two-layer neural networks, focusing on training only the last layer. In this particular configuration, the author initializes the first layer with a random binary vector and evaluates both mean-squared error and hinge loss as the loss functions. The authors assert that they have enhanced sample complexity while decreasing the number of iterations.

**Strengths:**

The author conducted a thorough analysis of network training for Boolean functions.

**Weaknesses:**

1. The organization of the paper is lacking, necessitating substantial restructuring.
- The problem setup demands a more detailed explanation. For instance, the fact that only the last layer is trained should be explicitly mentioned.
- Instead of delving into the proofs of Facts 3.2 and 3.3, it would be beneficial to provide a broader understanding of the main theorem, specifically Example 4.9, 4.10, and Theorem 4.11.

2. The notations used are ambiguous and inconsistent.
- The definition of $\sigma_1$ on page 3 is unclear.
- The same symbol, $u$, has been used for both weight and the vectorized function $u(f(T))$ on page 7.
- $W^k$ has not been clearly defined in Fact 3.2-3.
- The influence of the majority function is inadequately described in Fact 3.3.
- $f_i$ and $a_i$ are ambiguously defined on page 5.
- The symbol $\mathcal{J}$ is redundantly defined multiple times.
- There is no clear definition of $\mathcal{H}$.
- The best degree $k$ approximation needs a more meticulous definition.
- Neither definition 1.5 nor Theorem 1.6 is provided on page 7.

3. The paper would benefit from a more comprehensive discussion and comparison with the work of Barak et al. (2022) to highlight the - unique contributions of the present study.

4. The usage of contractions like "don't" and "won't" is not suitable for formal writing. It is recommended to avoid such informal language.

**Questions:**

Please check Weakness section.

---

### Official Review · Reviewer_cvKh · 2023-11-01

**Soundness:** 3 good
**Presentation:** 2 fair
**Contribution:** 2 fair
**Rating:** 3
**Confidence:** 3

**Summary:**

This paper studies the problem of learning Boolean functions and particularly parities with neural networks. More specifically, they consider a random features model trained with $\ell_2$ or hinge loss and they show that parities in dimension $n$ with degree $k$ can be learned with error $\epsilon$ assuming that the model has $n^{O(k)}poly(\frac{1}{\epsilon})$ samples and features and $n^{O(k)}\log(\frac{1}{\epsilon})$ iterations.

The paper further shows that in the provided training setting, lower degree monomials are learned faster than higher degree monomials.

**Strengths:**

- The result showing that lower degree features are learned faster than higher degree features is interesting, although the proof is limited to the proposed training setting.
- The learning results are applicable to many activation functions.

**Weaknesses:**

- In the abstract, the paper says that previous analyses of GD use the convexity of the problem while most loss functions are highly non-convex. Indeed, in this paper the loss functions considered are $\ell_2$ and hinge that are both convex. Further, the model is random features which makes the training linear and convex. (The paper also shows the convergence by almost proving a descent lemma.)
- Similarly, the paper mentions that the current theoretical results for Boolean functions are not close to the practice. While, in this paper, the training is on random features model with a large batch size. I personally think this method is even further from practice (compared to the works that employ layer-wise training for example). Note that in previous years, there have been many results showing the superiority of feature learning regimes over training in the kernel regime (including random features model).

**Questions:**

- Q1. Can the asymptomatics be made more precise? For example, the current version of the paper presents them as $n^{O(k)}$, but can it be made more exact, e.g., $O(n^{2k})$?
- Q2. How does the asymptotics (e.g., number of samples/features/iterations) compare to other results for kernel methods and other recent works such as [1]? For example, [1] discusses how one can compensate for a low number of iterations/samples/parameters by keeping the two others high.
- Q3. In the last paragraph of page 13, one has to prove $\frac{\zeta_k^2}{\zeta_l^2}= \Omega(n^{l-k})$. From fact 3.2, we have a lower bound of $\zeta_k \geq n^{-O(k)}$, but what is the upper bound for $\zeta_l$?


[1] Pareto Frontiers in Neural Feature Learning: Data, Compute, Width, and Luck -- https://arxiv.org/pdf/2309.03800.pdf

---

### Official Review · Reviewer_gUqw · 2023-11-07

**Soundness:** 2 fair
**Presentation:** 1 poor
**Contribution:** 3 good
**Rating:** 3
**Confidence:** 3

**Summary:**

This paper analyzes the learnability of Boolean functions using neural nets with a single hidden layer, and Relu activations when trained with gradient/hinge loss using gradient descent. They show that it performs fairly well learning parities of size $k$ in time close to the SQ lower bound of $n^\Omega(k)$. For arbitrary functions, they can prove learnability as good as the best low degree approximation, assuming certain spectral properties of  the transfer function.

**Strengths:**

1. I think the problem they study is interesting and important. Boolean functions are a natural setting where one can hope to rigorously analyze the kind of behavior that is empirically seen to hold for neural nests (like simple features are composed to get more complex ones).

2. The results seem technically quite solid. While I have not verified the details, it seems like the push the envelope of Boolean Fourier analysis by developing tools to analyze transfer functions like Relu. This might turn out to be important in the long run, encouraging more experts in this area to think about DNN-related problems.

3. The results themselves are interesting and solid  if not too  surprising.

**Weaknesses:**

The main weakness of the paper is its presentation, which I found very hard to penetrate (and I have worked in this area before). The notation is excessively heavy, the flow of results is not clear, a lot of heavy math is typeset inline in definitions and theorem statements. . It appears to me that the paper was written in a hurry. In its current form, it is inaccessible even to experts, leave alone non-experts.I feel the results are important enough that the paper ought to be substantially rewritten, and resubmitted (to a top venue). I cannot recommend acceptance in the current form.

**Questions:**

I will highlight just a couple of examples:

1. In defining $h^{\leq k}_{even}$ you say the best approximation with even degree and degree 1 support. Can you state more precisely what this means? Presumably if $k$ is even, this is just the level $k$ truncation? By even degree, I assume you mean that the highest degree term is even? In which case, why do we need to specify "Degree 1 support" separately? Or do you mean projecting the spectrum onto the even weight subsets, while also keeping the degree one terms?

2. It might be worth highlighting that $J(g)$ is a convolution operator. Isn't Proposition 4.1 just a consequence of the standard fact that convolution in the time domain is pointwise multiplication in the Fourier domain?

3. Remark 4.8 claims that Def 4.7 is "equivalent to: there is some $u$ so that $|\hat{\sigma}_u(S)| \geq \alpha $ or any $s \in S$. This statement does not parse. Perhaps you meant "for any $s \in S$ and not "or"? That statement does parse, but it is not equivalent to Def 4.7 which requires the condition to hold for all $u$.

4. There are too many theorem statements in section 4, and their organization could perhaps be better. The one line proofs for Theorem 4.3, 4.4 etc are not very helpful to the reader to understand what is going on. If the main results are Theorem 4.11 and 4.12 then one might state those first, give some intuiton for their proofs, and derive the other results as corollaries of these results.

---

### Meta-Review · Area_Chair_UmJF · 2023-12-12

**Metareview:**

The paper provides a theoretical analysis of learning Boolean functions with neural networks and gradient descent. The authors study learning on a single hidden layer and ReLU activations, under two different loss functions. The actual updates are applied to the final layer weights only. The authors obtain guarantees on sample complexity and number of iterations needed to learn within some precision parities of a fixed dimension and degree. As currently written, the paper is incredibly hard to follow partly due to lack of mathematical clarity; mathematical clarity is a necessity for a theory paper. As stated by one of the reviewers, the paper is barely accessible even to experts in the field. The results are quite limited: while the abstract states that training of a 2 layer neural network is being studied, only the final layer is trained, and the loss functions considered are convex, thus collapsing back to the convex training regime. Therefore, an argument made in the abstract and the paper that this study goes beyond the convexity assumptions made in previous work is not applicable, and even misleading.

**Justification For Why Not Higher Score:**

Presentation and readability of mathematical arguments needs to be heavily improved. The authors need to rethink they convexity argument and faithfully present their contributions relative to prior work. The paper should also improve interpretation and analysis of the results.

**Justification For Why Not Lower Score:**

N/A

---

### Decision · Program_Chairs · 2024-01-16

Reject